# GEOMETRY MATTERS: EXPLORING LANGUAGE EXAMPLES AT THE DECISION BOUNDARY

## ABSTRACT

A growing body of recent evidence has highlighted the limitations of natural language processing (NLP) datasets and classifiers. These include the presence of annotation artifacts in datasets, classifiers relying on shallow features like a single word (e.g., if a movie review has the word "romantic", the review tends to be positive), or unnecessary words (e.g., learning a proper noun to classify a movie as positive or negative). The presence of such artifacts has subsequently led to the development of challenging datasets to force the model to generalize better. While a variety of heuristic strategies, such as counterfactual examples and contrast sets, have been proposed, the theoretical justification about what makes these examples difficult for the classifier is often lacking or unclear. In this paper, using tools from information geometry, we propose a theoretical way to quantify the difficulty of an example in NLP. Using our approach, we explore difficult examples for several deep learning architectures. We discover that both BERT, CNN and fasttext are susceptible to word substitutions in high difficulty examples. These classifiers tend to perform poorly on the FIM test set. (generated by sampling and perturbing difficult examples, with accuracy dropping below 50%). We replicate our experiments on 5 NLP datasets (YelpReviewPolarity, AGNEWS, SogouNews, YelpReviewFull and Yahoo Answers). On YelpReviewPolarity we observe a correlation coefficient of -0.4 between resilience to perturbations and the difficulty score. Similarly we observe a correlation of 0.35 between the difficulty score and the empirical success probability of random substitutions. Our approach is simple, architecture agnostic and can be used to study the fragilities of text classification models. All the code used will be made publicly available, including a tool to explore the difficult examples for other datasets.

## 1 INTRODUCTION

Machine learning classifiers have achieved state-of-the-art success in tasks such as image classification and text classification. Despite their successes, several recent papers have pointed out flaws in the features learned by such classifiers. Geirhos et al. (2020) cast this phenomenon as shortcut learning, where a classifier ends up relying on shallow features in benchmark datasets that do not generalize well to more difficult datasets or tasks. For instance, Beery et al. (2018) showed that an image dataset constructed for animal detection and classification failed to generalize to images of animals in new locations. In language, this problem manifests at the word level. Poliak et al. (2018) showed that models using one of the two input sentences for semantic entailment performed better than the majority class by relying on shallow features. Similar observations were also made by Gururangan et al. (2018), where linguistic traits such as "vagueness" and "negation" were highly correlated with certain classes.

In order to study the robustness of a classifier, it is essential to perturb the examples at the classifier's decision boundary. Contrast sets by Gardner et al. (2020) and counterfactual examples by Kaushik et al. (2020) are two approaches where the authors aimed at perturbing the datasets to identify difficult examples. In contrast sets, authors of the dataset manually fill in the examples near the decision boundary (examples highlighted in small circles in Figure 1) to better evaluate the classifier performance. In counterfactual examples, the authors use counterfactual reasoning along with Amazon Mechanical Turk to create the "non-gratuitous changes." While these approaches are interesting, it's still unclear if evaluating on these will actually capture a classifier's fragility. Furthermore, these

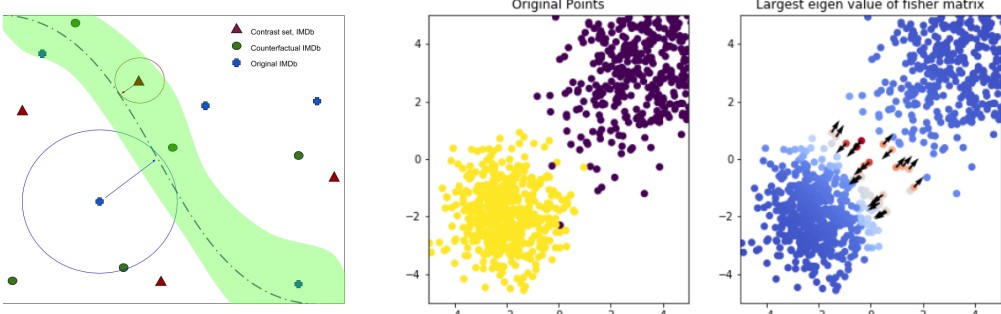

Figure 1: Quantifying difficulty by using the largest eigenvalue of the Fisher information metric (FIM) a) We show that contrast sets and counterfactual examples aren't necessarily concentrated near the decision boundary as shown in this diagram. Difficult examples are the ones shown in green region (close to the decision boundary) and this is the region where we should evaluate model fragility. b) We sample points from a two-component Gaussian mixture model. We next train a classifier to separate the two classes. c) Dataset colored by the eigenvalue of the FIM, difficult examples with a higher eigenvalue lie closer to the decision boundary.

approaches significantly differ from each other and it's important to come up with a common way to reason about them.

Motivated by these challenges, we propose a geometrical framework to reason about difficult examples. Using our method, we are able to discover fragile examples for state of the art NLP models like BERTby Devlin et al. (2018) and CNN (Convolutional Neural Networks) by Kim (2014). Our experiments using the Fisher information metric (FIM) show that both counterfactual examples and contrast sets are, in fact, quite far from the decision boundary geometrically and not that different from normal examples (circles and triangles in Figure 1). As such, it is more important to perform evaluation on the examples lying in the green region, which represent confusing examples for the classifier, where even a small perturbation (for instance, substituting the name of an actress) can cause the neural network to misclassify. It is important to note that this does not depend solely on the classifier's certainty as adversarial examples can fool neural networks into misclassifying with high confidence, as was shown by Szegedy et al. (2013).

We now motivate our choice of using the Fisher information metric (FIM) in order to quantify the difficulty of an example. In most natural language processing tasks, deep learning models are used to model the conditional probability distribution $p(y \mid x)$ of a class label $y$ conditioned on the input $x$. Here $x$ can represent a sentence, while $y$ can be a sentiment of the sentence. If we imagine a neural network as a probabilistic mapping between inputs to outputs, a natural property to measure is the Kullback-Leibler (KL) divergence between the example and an $\epsilon$ perturbation around that example. For small perturbations to the input, the FIM gives a quadratic form that approximates, up to second order, the change in the output probabilities of a neural network. Zhao et al. (2019) used this fact to demonstrate that the eigenvector associated with the maximum eigenvalue of the FIM gives an effective direction to perturb an example to generate an adversarial attack in computer vision. Furthermore, from an information geometry viewpoint, the FIM is a Riemannian metric, inducing a manifold geometry on the input space and providing a notion of distance based on changes in the information of inputs. To the best of our knowledge, this is the first work analyzing properties of the fisher metric to understand classifier fragility in NLP.

The rest of the paper is organized as follows: In Section 2, we summarize related work. In Section 3, we discuss our approach of computing the FIM and the gradient-based perturbation strategy. In Section 4, we discuss the results of the eigenvalues of FIM in synthetic data and sentiment analysis datasets with BERT and CNN. Finally, in Section 5, we discuss the implications of studying the eigenvalues of FIM for evaluating NLP models.

Table 1: CNN, IMDb dataset: Unlike the difficult examples (larger eigenvalue), word substitutions are ineffective in changing the classifier output for the easier examples (smaller eigenvalue). In difficult examples synonym or change of name, changes classifier label. In easy examples, despite multiple simultaneous antonym substitutions, the classifier sentiment does not change.

| Perturbed sentiment | Word substitutions |
|---|---|
| **Positive** → **Negative** difficult example ($\lambda_{max}$ =**5.25**) | Going into this movie, I had heard good things about it. Coming out of it, I wasn't really amazed nor disappointed. Simon Pegg plays a rather childish character much like his other movies. There were a couple of laughs here and there– nothing too funny. Probably my **favorite** → **preferred** parts of the movie is when he dances in the club scene. I totally gotta try that out next time I find myself in a club. A couple of stars here and there including: Megan Fox, Kirsten Dunst, that chick from X-Files, and Jeff Bridges. I found it quite amusing to see a cameo appearance of Thandie Newton in a scene. She of course being in a previous movie with Simon Pegg, Run Fatboy Run. I see it as a toss up, you'll either enjoy it to an extent or find it a little dull. I might add, **Kirsten Dunst** → **Nicole Kidman, Emma Stone, Megan Fox, Tom Cruise, Johnny Depp, Robert Downey Jr.** is adorable in this movie. :3 |
| **Negative** → **Negative** easy example ($\lambda_{max}$ =**0.0008**) | I missed this movie in the cinema but had some idea in the back of my head that it was worth a look, so when I saw it on the shelves in DVD I thought "time to watch it". Big mistake!

A long list of stars cannot save this turkey, surely one of the **worst** → **best** movies ever. An **incomprehensible** → **comprehensible** plot is **poorly** → **exceptionally** delivered and **poorly** → **brilliantly** presented. Perhaps it would have made more sense if I'd read Robbins' novel but unless the film is completely different to the novel, and with Robbins assisting in the screenplay I doubt it, the novel would have to be an **excruciating** → **exciting** read as well.

I hope the actors were well paid as they looked embarrassed to be in this waste of celluloid and more lately DVD blanks, take for example Pat Morita. Even Thurman has the grace to look uncomfortable at times.

Save yourself around 98 minutes of your life for something more worthwhile, like trimming your toenails or sorting out your sock drawer. Even when you see it in the "under $5" throw-away bin at your local store, resist the urge! |

# 2 RELATED WORK

In NLP, machine learning models for classification rely on spurious statistical patterns of the text and use *shortcut* for learning to classify. These can range from annotation artifacts, as was shown by Goyal et al. (2017); Kaushik and Lipton (2018); Gururangan et al. (2018), spelling mistakes as in McCoy et al. (2019), or new test conditions that require world knowledge Glockner et al. (2018). Simple decision rules that the model relies on are hard to quantify. Trivial patterns like relying on the answer "2" for answering questions of the format "how many" for the visual question answering dataset Antol et al. (2015), would correctly answer 39% of the questions. Jia and Liang (2017) showed that adversarially inserted sentences that did not change the correct answer, would cause state of the art models to regress in performance in the SQuAD Rajpurkar et al. (2016) question answering dataset. Glockner et al. (2018) showed that template-based modifications by swapping just one word from the training set to create a test set highlighted models' failure to capture many simple inferences. Dixon et al. (2018) evaluated text classifiers using a synthetic test set to understand unintended biases and statistical patterns. Using a standard set of demographic identity terms, the authors reduce the unintended bias without hurting the model performance. Shen et al. showed that word substitution strategies include stylistic variations that change the sentiment analysis algorithms for similar word pairs. Evaluations of these models through perturbations of the input sentence are crucial to evaluating the robustness of models.

Another issue of language recently has been that static benchmarks like GLUE by Wang et al. (2018) tend to saturate quickly because of the availability of ever-increasing compute and harder benchmarks are needed like SuperGlue by Wang et al. (2019). A more sustainable approach to this is the development of moving benchmarks, and one notable initiative in this area is the Adversarial NLI by Nie et al. (2019), but most of the research community hardly validate their approach against this sort of moving benchmark. In the Adversarial NLI dataset, the authors propose an iterative, adversarial human-and-model-in-the-loop solution for Natural Language Understanding dataset collection, where the goal post continuously shifts about useful benchmarks and makes models robust by training the model iteratively on difficult examples. Approaches like never-ending learning byMitchell et al. (2018) where models improve, and test sets get difficult over time is critical. A moving benchmark is necessary since we know that improving performance on a constant test set may not generalize to newly collected datasets under the same condition Recht et al. (2019); Beery et al. (2018). Therefore, it is essential to find difficult examples in a more disciplined way.

Approaches based on geometry have recently started gaining traction in computer vision literature. Zhao et al. (2019) et al used a similar approach for understanding adversarial examples in images.

## 3 METHODS

A neural network with discrete output can be thought of as a mapping between a manifold of inputs to the discrete output space. Most traditional formulations treat this input space as flat, thus reasoning that the gradient of the likelihood in input space gives us the direction which causes the most significant change in terms of likelihood. However, by imagining the input as a pullback of the output, we obtain a non-linear manifold where the euclidean metric no longer suffices. A more appropriate choice thus is to use the fisher information as a Riemannian metric tensor.

We first introduce the Fisher Metric formulation for language. For the purposes of the derivation below the following notations are used.

$x$ : Vector of input sentence. This is an n * d sentence where n is the number of words in the sentence and d is the dimensionality of the word embedding.
$y$ : Label of class, in our context that is the positive or the negative sentiment.
**p(y|x)** : The conditional probability distribution between y and x.
$KL(p, q)$ : The KL divergence between distributions p and q for two sentences
$\nabla f(x)$ : Gradient of a function of f w.r.t x
$\nabla^2 f(x)$ : Hessian of f(x) w.r.t x

We apply the a perturbation $\eta$ to modifying a sentence to create a new sentence (eg., a counterfactual example). We can then see the effect of this perturbation $\eta$ in terms of change in the probability distribution over labels. Ideally, we would like to find points where a small perturbation can result in a large change in the probability distribution over labels.

$$KL(p(y|x)||p(y|x+\eta)) = -E_{p(y|x)}logp(y|x) + E_{p(y|x)}logp(y|x+\eta)$$

We now perform a Taylor expansion of the first term on the right hand side

$$= -E_{p(y|x)}(logp(y|x) + \eta\nabla logp(y|x) + \eta^T\nabla^2 logp(y|x)\eta + ...) + E_{p(y|x)}logp(y|x)$$

$$\sim -E_{p(y|x)}\eta^T\nabla^2 logp(y|x)\eta$$

Since the expectation of score is zero and the first and last terms cancel out, we are left with.

$$= \eta^T G\eta$$

Where G is the FIM. By studying the eigenvalues of this matrix locally, we can quantify if small change in $\eta$ can cause a large change in the distribution over labels. We use the largest eigenvalue of the FIM as a score to quantify the "difficulty" of an example. We now propose the following algorithm to compute the FIM:

After getting the eigenvalues of the FIM, we can use the largest eigenvalue $\lambda_{max}$ to quantify how fragile an example is to linguistic perturbation. At points with largest eigenvalues, smaller perturbations can be much more effective in changing the classifier output. These examples, thus, are also more confusing and more difficult for the model to classify.

Table 2: CNN, Counterfactual Examples: In difficult examples (larger eigenvalue), individual synonym/antonym substitutions are effective in changing the classifier output. In easy examples (smaller eigenvalue) multiple antonym substitutions simultaneously have no effect on the classifier output.

| Perturbed sentiment | Word substitutions |
| --- | --- |
| Positive → Negative difficult example $\lambda_{max}$ =4.38 | This move was on TV last night. I guess as a time filler, because it was incredible! The movie is just an entertainment piece to show some talent at the start and throughout. (Not **bad** talent at all). But the story is too brilliant for words. The "wolf", if that is what you can call it, is hardly shown fully save his teeth. When it is fully in view, you can clearly see they had some interns working on the CGI, because the wolf runs like he's running in a treadmill, and the CGI fur looks like it's been waxed, all shiny :)

The movie is full of gore and blood, and you can hardly spot who is going to get killed/slashed/eaten next. Even if you like these kind of splatter movies you will be surprised, they did do a good job at it.

Don't even get me started on the actors... Very **amazing** lines and the girls hardly scream at anything. But then again, if someone asked me to do good acting just to give me a few bucks, then hey, where do I sign up?

Overall **exciting** → **boring, extraordinary, uninteresting, exceptional** and frightening horror. |
| Negative → Negative easy example $\lambda_{max}$ =0.013 | I couldn't stand this movie. It is a definite **waste** of a movie. It fills you with boredom. This movie is not worth the rental or worth buying. It should be in everyones trash. **Worst** → **Excellent** movie I have seen in a long time. It will make you mad because everyone is so mean to Carl Brashear, but in the end it gets only worse. It is a story of **cheesy** romance, **bad** → **good** drama, action, and plenty of **unfunny** → **funny** lines to keep you rolling your eyes. I hated a lot of the quotes. I use them all the time in mocking the film. They did not help keep me on task of what I want to do. It shows that anyone can achieve their dreams, all they have to do is whine about it until they get their way. It is a long movie, but every time I watch it, I dradr that it is as long as it is. I get so bored in it, that it goes so slow. I hated this movie. I never want to watch it again. |

---

**Algorithm 1** Algorithm for estimating difficulty of an example

**Input:** x, f
**Output:** $\lambda_{max}$
   *Calculate probability vector :*
 1: $p = f(x)$
   *Calculate Jacobian of log probability w.r.t x*
 2: $J = \nabla_x log p$
   *Duplicate probability vector along rows to match J's shape*
 3: $p_c = duplicate(p, J.dim[0])$
   *Compute the FIM*
 4: $G = p_c J J^T$
   *Perform eigendecomposition to get the eigenvalues*
 5: $\lambda_s, v_s = eigendecomposition(G)$
 6: **return** $max(\lambda_s)$

---

### 3.1 GRADIENT ATTRIBUTION BASED PERTURBATION

In our perturbations, we rely on Integrated Gradients (IG) by Sundararajan et al. (2017), since IG satisfies both sensitivity (network gradients to focus on relevant input feature attributes of the model with respect to the output) and implementation invariance of the neural network. IG allows us to assign relative importance to each word token in the input. After computing the token attributions with IG, we only perturb those words for "easy" and "difficult" examples. Once we compute the feature attribution for each input word in the sentence, we find the most important words by thresholding the attribution score. We then test the classifier's fragility by substituting these words for their synonyms/antonyms, etc, and checking if the classifier prediction changes. We show that easy examples are robust to significant edits with multiple positive tokens like "excellent", "great" substituted simultaneously, and "difficult" examples change predictions from positive to negative with meaningless substitutions like names of actors and actresses.

## 4 DISCUSSION AND RESULTS

### 4.1 FIM REFLECTS DISTANCES FROM THE DECISION BOUNDARY

We first investigate the FIM properties by training a neural network on a synthetic mixture of gaussians dataset. The parameters of the two gaussians are $\mu_1 = [-2, -2]$ and $\mu_2 = [3.5, 3.5]$. The covariances are $\Sigma_1 = eye(2)$ and $\Sigma_2 = [[2., 1.], [1., 2.]]$ The dataset is shown in figure 1. We train a 2-layered network to separate the two classes from each other. We use algorithm 1 to compute the largest

eigenvalue of the FIM for each datapoint and use it to color the points. We also plot the eigenvector for the top 20 points.

As seen by the gradient of the colors in Figure 1, the points with the largest eigenvalue of the FIM lie close to the decision boundary. These points are indicative of how confusing the example is to the neural network since a small shift along the eigenvector can cause a significant change in the KL divergence between the probability distribution of the original and new data points. These points with a high eigenvalue are close to the decision boundary, and these examples are most susceptible to perturbations.

## 4.2 FIM VALUES CAPTURE RESILIENCE TO LINGUISTIC PERTURBATIONS, QUALITATIVE EXPLORATION

### 4.2.1 CNN

For the difficult examples, we tried the following trivial substitutions one at a time: a) a synonym b) a semantically equivalent word c) an antonym d) substituting the name of an actress present in the same passage. As seen in Table 1, either replacing favorite with preferred or "Kirsten Dunst" with any of the listed actors/actresses suffices to change the classifier's prediction. Note that, "Megan Fox's" name appears in the same review in the previous sentence. Similar in Table 3, for counterfactual examples, it's sufficient to replace "exciting" with either an antonym ("boring" or "uninteresting") or a synonym ("extraordinary" or "exceptional"). We see the same pattern in contrast sets in Appendix.

For easy examples however, despite trying to replace four or more high attribution words simultaneously with antonyms, the predicted sentiment did not change. Substitutions include "good" to "bad", "unfunny" to "funny", "factually correct" to "factually incorrect". Even though the passage included words like "boredom", a word that is generally associated with a negative movie review, the model did not assign it a high attribution score. Consequently, we did not try to substitute these words for testing robustness or fragility of word substitutions.

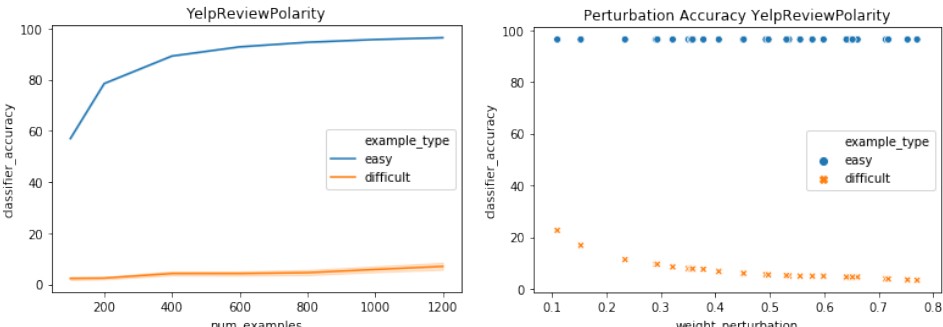

Figure 2: Left figure: The difficult examples on the FIM test set are challenging for the classifiers with accuracy between 1-5%. For the easy examples, (low FIM $\lambda_{max}$) the accuracy ranges between 60-100%. Right figure: We plot the classifier accuracy with respect to the l2 norm of the perturbation in embedding space. For difficult examples, the classifier accuracy drops below 5% at small perturbation strengths (l2 norm < 0.3)

### 4.2.2 BERT

BERT: Transfer learned models like BERT capture rich semantic structure of the sentence. They are robust to changes like actor names and tend to rely on semantically relevant words for classifying movie reviews. As we can see from Table 6 difficult BERT examples, even with multiple positive words, tend to predict a negative sentiment when only one of the positive word is substituted. Even with words like "fantastic", "terrific" and "exhilarating", just changing "best" to "worst" changed the entire sentiment of the movie review. Easy examples for BERT require multiple simultaneous word substitutions to change the sentiment as can be seen in Table 6. Unlike CNN models, BERT is significantly more robust to meaningless substitutions like actor names.

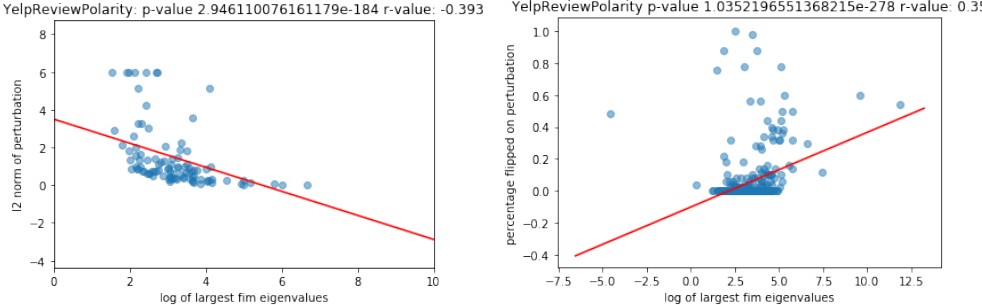

Figure 3: On YelpReviewPolarity we observe a correlation coefficient of -0.4 between minimum perturbation strength and the difficulty score. Similarly, we observe a correlation of 0.35 between the difficulty score and empirical success of random word substitutions. This suggests that fim eigenvalue captures perturbation sensitivity in both embedding space and word substitutions.

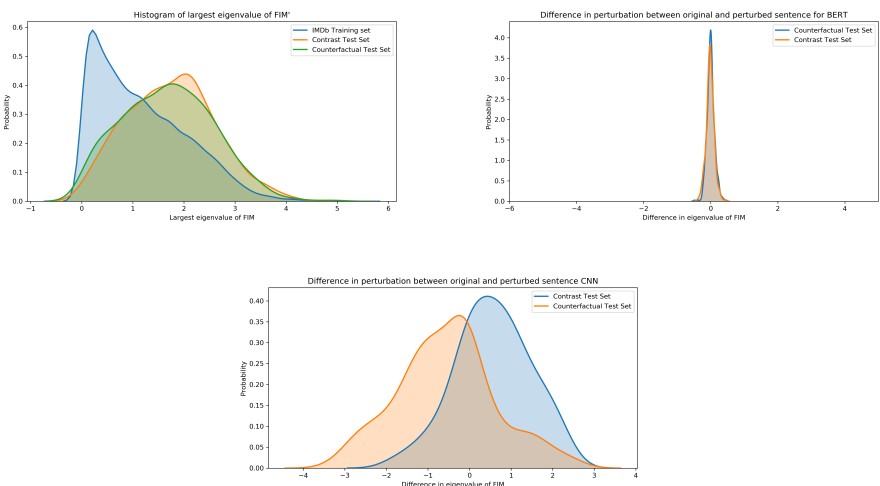

Figure 4: a) Distribution of difference in largest eigenvalue of FIM of the original and the perturbed sentence in contrast set and counterfactual examples for BERT and CNN. With a mean near 0, these perturbations are not difficult for the model. Adhoc perturbations are thus not useful for evaluating model robustness.

### 4.3 HIGH FIM EXAMPLES ARE CHALLENGING TO NLP CLASSIFIERS, QUANTITATIVE EXPERIMENTS

#### 4.3.1 BAG OF TRICKS FOR EFFICIENT TEXT CLASSIFICATION

When the dataset size is increased, simple models perform competitively to models like BERT Wang et al. (2020). We use the bag of tricks for efficient text classification Joulin et al. (2017) for its inference and training speed and strong performance on text classification on 5 popular text classification datasets (YelpReviewPolarity, AGNEWS, SogouNews, YelpReviewFull and Yahoo Answers)

In order to investigate the relationship between classifier performance and FIM eigenvalues we select n-smallest and n-largest fim examples from the test set. We randomly chose a perturbation strength between 0 and 1 and we perturb the example along the largest eigenvector proportional to the chosen perturbation strength. As can be seen from figure 2 left, despite sampling a large range of difficult examples, between (125-1200), the classifier performed poorly on the difficult set with the accuracy failing to exceed 9%. On the easy set however, the classifier accuracy never falls below

57.5%. In figure 2b, we plot the classifier accuracy as a function of perturbation strength (l2 norm of perturbation vector for difficult examples, even small perturbations upto 0.4 l2 norm, cause a 17% reduction in accuracy. For easy examples however even upto a l2 norm of 1 barely result in any drop in performance. More examples are in Appendix A.

We randomly sample 500 examples and perturb the examples along the largest eigenvector. We use binary search to determine the minimum perturbation strength sufficient to flip the classifier output. As we can see from Figure 3, a correlation of -0.4 implies it is possible to misclassify larger FIM examples by applying a small perturbation strength. In order to investigate the relationship between FIM and random word substitutions, we flipped 10% of the words in a document, for 500 randomly sampled examples. As seen in Figure 3b, a correlation of 0.35 implies that difficult examples, (larger FIM eigenvalues) are more susceptible to random word substitutions.

### 4.4 Do contrast sets and counterfactual examples lie close to the decision boundary?

We first plot the distribution of eigenvalues of IMDb examples, counterfactual examples and contrast sets examples in Figure 2. If the goal is to evaluate examples near the decision boundary the contrast set/counterfactual eigenvalue distribution should be shifted to the right with very little overlap with the original examples. However, as we can see from Figure 2, we see a 69.38% overlap between contrast sets and IMDb examples as well as a 73.58% overlap between counterfactual and IMDb examples for the CNN model that was only trained on IMDb training set. The presence of significant overlap between the three distributions indicate that counterfactual/contrast examples are not more difficult for the model compared to the normal examples. Furthermore, since the FIM capture distance from the decision boundary, most counterfactual and contrast sets lie as far away from the decision boundary as the original IMDb dataset examples.

We next quantify the effect of changing the original sentence to the perturbed sentence in counterfactual/contrast test sets. We plot the distribution of difference in eigenvalue of the original and perturbed sentence in Figure 2. 30% of the counterfactual examples (area to the right of 0) did not increase the FIM eigenvalue and the perturbation was ineffective. For contrast sets, this number is around 70% (area to the right of 0), hinting that majority of perturbations are not useful for testing model robustness. A subtle point to note here, even though 70% of counterfactual examples increase the difficulty, it's because we have chosen a weak threshold (0) to quantify the usefulness of the perturbation. A more practical threshold like 1 would lead to lesser number of useful examples for both counterfactual and contrast set examples.

## 5 Implications

### 5.1 Constructing Test set by perturbing high FIM examples

As evident from figure 2, high FIM examples are particularly challenging for NLP classifiers. This makes them useful to understand model performance. By sampling the high FIM examples and perturbing them slightly in embedding space, we can construct a new test set to evaluate NLP models. By repeating this process multiple times, more robust classifiers can be created.

### 5.2 NLP models should be evaluated at the examples near the decision boundary

Deep learning models because of their high representation capacity are good at memorizing the training set. In the absence of sufficient variation in the test set, this can lead to an inflation in accuracy without actual generalization. However, examples near the decision boundary of a classifier, show the fragilities of these classifiers. These are the examples that are most susceptible to shallow feature learning and thus are the ones that need to be tested for fragility and word substitutions. Our approach based on FIM score, provides a task and architecture agnostic approach to discovering such examples.

Table 3: Statistics of difference in largest FIM eigenvalue (pre and post perturbation) for counterfactual examples and contrast sets.

| Architecture | Dataset | Type | Mean | Std |
|---|---|---|---|---|
| CNN | Counterfactual Examples | Dev | -0.55 | 1.17 |
| | | Test | -0.57 | 1.22 |
| CNN | Contrast Sets | Dev | 0.56 | 0.89 |
| | | Test | 0.65 | 1.15 |
| BERT | Counterfactual Examples | Dev | 0.004 | 0.10 |
| | | Test | 0.003 | 0.11 |
| BERT | Contrast Sets | Dev | 0.02 | 0.10 |
| | | Test | -0.01 | 0.11 |

### 5.3 INVESTIGATE HIGH FIM EXAMPLES WITH INTEGRATED GRADIENTS TO UNDERSTAND MODEL FRAGILITY

For our models, difficult examples have a mix of positive and negative words in a movie review. The models also struggled with examples of movies that selectively praise some attributes like acting (e.g., "Exceptional performance of the actors got me hooked to the movie from the beginning") while simultaneously use negative phrases (e.g., "however the editing was horrible"). Difficult examples also have high token attributions associated with irrelevant words like "nuclear," "get," and "an." Thus substituting one or two words in difficult examples change the predicted label of the classifier. Similarly, easier examples have clearly positive reviews (e.g., "Excellent direction, clever plot and gripping story"). Combining integrated gradients with high FIM examples can thus yield insights into the fragility of NLP models.

### 5.4 PERTURBATION SHOULD BE PERFORMED ALONG THE LARGEST EIGENVECTOR

The eigenvector corresponding to $\lambda_{max}$ represents the direction of largest change in probability distribution over output. This makes it a useful candidate direction to evaluate model fragility while random word substitutions are easier to interpret, they might end up being orthogonal to the largest eigenvector and thus not capture model fragility effectively. One strategy to circumvent this issue is to find word substitutions with a large projection on this eigenvector.

## 6 CONCLUSION

We have proposed a geometrical method to quantify the difficulty of an example in NLP. Our method identified fragilities in state of the art NLP models like BERT and CNN. By directly modeling the impact of perturbation in natural language through the Fisher information metric, we showed that examples close to the decision boundary are sensitive to meaningless changes. We also showed that counterfactual examples and contrast sets don't necessarily lie close to the decision boundary. Furthermore, depending on the distance from the decision boundary, small innocuous tweaks to a sentence might actually correspond to a large change in the embedding space. Thus one has to be careful in constructing perturbations, and more disciplined approaches are needed to address the same.

As our methods are agnostic to the choice of architecture or dataset, in future we plan to extend this to other NLP tasks and datasets. We are also studying the properties of the decision boundary: whether different classifiers have a similar decision boundary, are there universal difficult examples which confound multiple classifiers? We also plan to investigate strategies for automatic generation of sentence perturbations based on the largest eigenvector of the FIM. Even though we have explored difficult examples from a classifier's perspective, we are also interested in exploring the connections between FIM and perceived difficulty of examples by humans.

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

## APPENDIX A

CNN and BERT:

We train a convolutional neural network (CNN) with a 50d GloVe embedding on the IMDb dataset, and calculate the eigenvalue of the FIM for each example. The accuracy on the IMDb test set was around 85.4%. For all experiments in the paper the model was trained on the original 25000 examples in the original IMDb training split with a 90% train and 10% valid split. For all experiments in the paper, we did not train on the counterfactual or contrast set examples to fairly evaluate the robustness to perturbations of contrast sets and counterfactual examples. For BERT we finetuned 'bert-base-uncased' and achieved an accuracy of 92.6% using huggingface transformers by Wolf et al. (2019). We evaluated the largest eigenvalue of the dev and test sets of the contrast set and counterfactual examples datasets of IMDb.

Bag of Tricks for efficient text classification:

We use unigram and bigram word embeddings. The latent dimension is 32. We use torchtext datasets directly available for the training and test splits. We use the SGD optimizer with a learning rate of 4 and train each model for 8 epochs. We use a learning rate scheduler with a gamma of 0.9.

# APPENDIX B

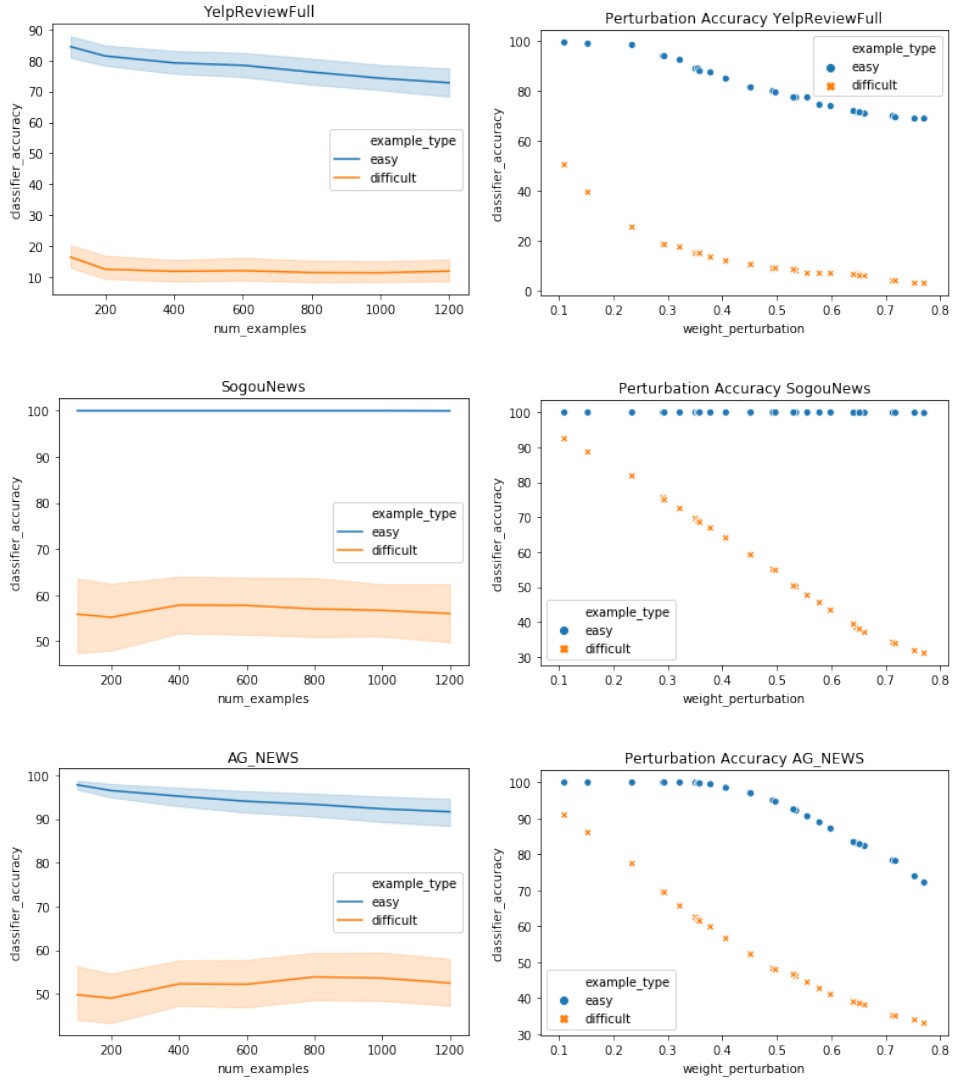

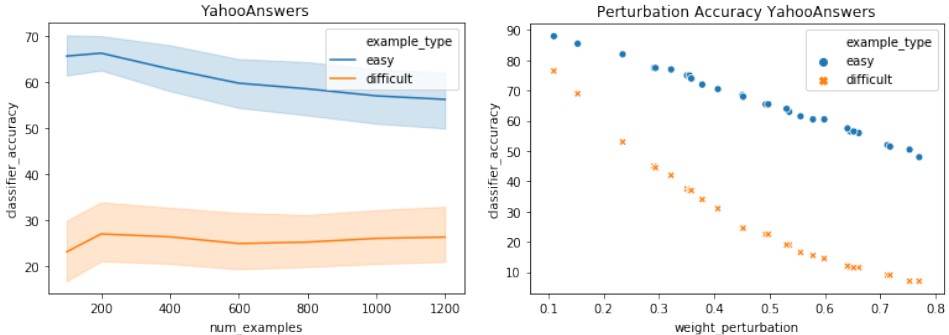

Figure 5: Here we take the Joulin et al. (2017) text classification approach ( and plot classifier accuracy as a function of the step size of the perturbation. The number of examples in the x-axis represents the number of examples based on the eigenvalue. So, 200 refers to the 200 easy examples and 200 difficult examples. We then perturb these examples in the direction of the eigenvector and check if the classifier prediction flipped. As can be seen that the classifier accuracy remains very high for easy examples and is significantly low for difficult examples. On the second diagram, for easy examples, the classifier still exhibits minimal performance drop when the weight of the perturbation along the eigenvector is increased. Thus our method is dataset agnostic.

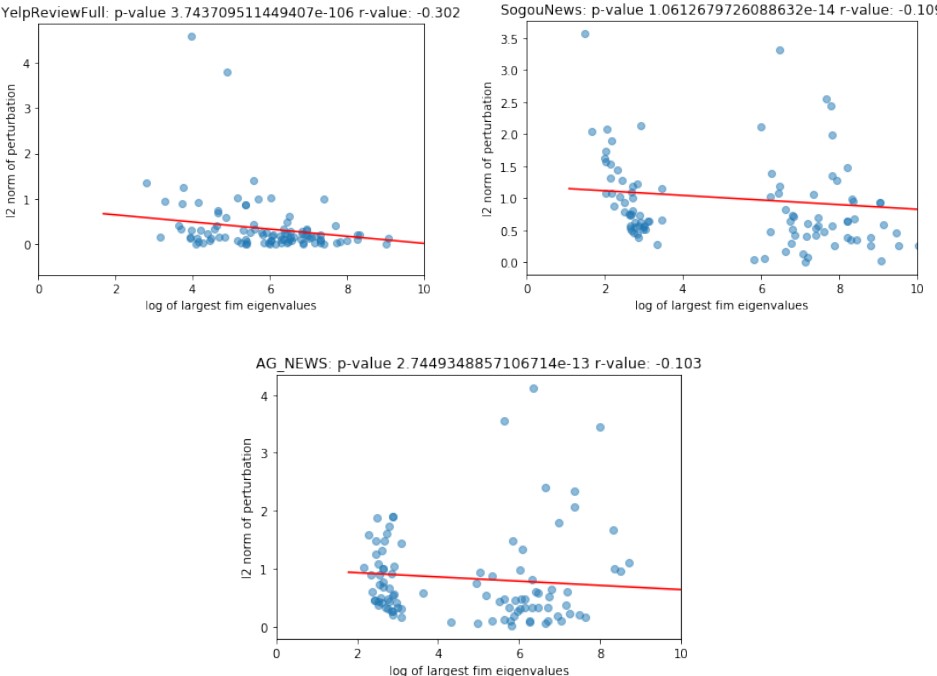

Figure 6: We use the eigenvector with the largest eigenvalue as a perturbation in embedding space. As we can see there is a linear relationship between log lambda max and the perturbation strength needed to flip the classifier output. We use binary search between the range (0,6) to discover the minimal l2 norm perturbation along the largest eigenvector which can successfully flip the classifier's output. For each dataset, we select 500 random examples from the test set for the perturbation. Note the negative slope in all the datasets. The p-value and the r-value are reported at the top. The log of the largest fim eigenvalue is on the x-axis. Thus, an example with a small eigenvalue requires a larger perturbation to flip classifier prediction than an example with large fim eigenvalue.

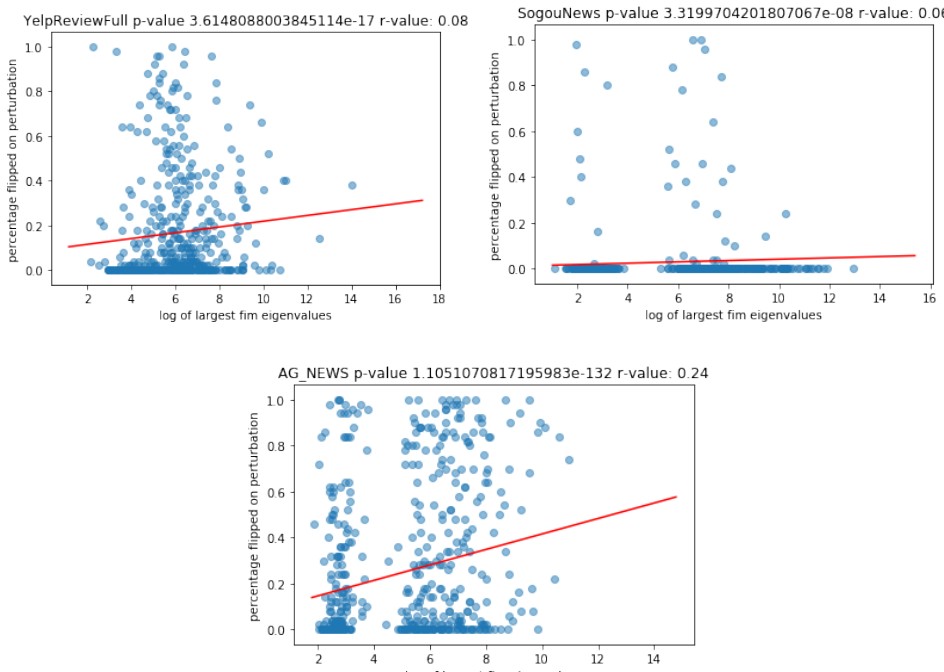

Figure 7: We notice the linear relationship between the log of eigenvalue and the percentage of successful word flips. Since the length of each document varies drastically for these document classification problems, we decide the number of words to flip as $10\%$ of the document length with the vocabulary of the dataset. We then measure the percentage of predictions whose classification label changes and the log of fim eigenvalue. The $p_value and the r_value are reported at the top$.

APPENDIX C

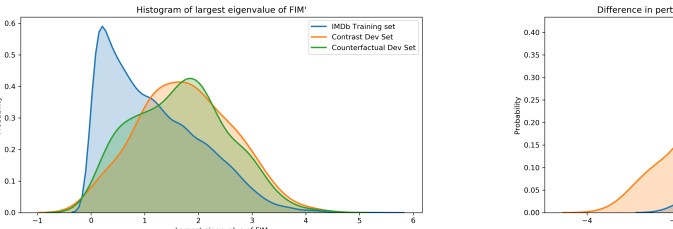

Figure 8: a) Histogram of eigenvalues of dev sets of original IMDb examples, contrast sets and counterfactual examples. The significant overlap between the three distributions indicates that the counterfactual and contrast set examples might not be as difficult as previously believed. The tail end of all three distributions contain the difficult examples. b) Distribution of difference in largest eigenvalue of FIM of the original and the perturbed sentence in contrast set and counterfactual examples. With a mean near 0, these perturbations are not necessarily more difficult for the model.

Table 4: Perturbing an easy example on the IMDb dataset: The first row represents original sentence. As we can see here, most perturbations are ineffective in changing the FIM eigenvalue and thus the difficulty of the example. Despite multiple substitutions (last row), we are only able to achieve a modest increase in FIM score, indicating the resilience to perturbations.

| Perturbed sentiment | Word substitutions |
|---|---|
| **Negative** → **Negative** easy example ($\lambda_{max} =$**0.0007**) | Probably the **worst** Dolph film ever. There's nothing you'd want or expect here. Don't **waste** your time. Dolph plays a **miserable** cop with no interests in life. His brother gets killed and Dolph tries to figure things out. The character is just plain **stupid** and stumbles around aimlessly. Pointless. |
| **Negative** → **Negative** minor change in FIM value ($\lambda_{max} =$**0.0008**) | Probably the worst Dolph film ever. There's **nothing** → **everything** you'd want or expect here. Don't waste your time. Dolph plays a miserable cop with no interests in life. His brother gets killed and Dolph tries to figure things out. The character is just plain stupid and stumbles around aimlessly. Pointless. |
| **Negative** → **Negative** minor change in FIM value ($\lambda_{max} =$**0.0005**) | Probably the worst Dolph film ever. There's nothing you'd want or expect here. Don't waste your time. Dolph **plays** → **portrays** a miserable cop with no interests in life. His brother gets killed and Dolph tries to figure things out. The character is just plain stupid and stumbles around aimlessly. Pointless. |
| **Negative** → **Negative** minor change in FIM value ($\lambda_{max} =$**0.0007**) | Probably the worst Dolph film ever. There's nothing you'd want or expect here. Don't waste your time. Dolph **plays** → **portrays** a miserable cop with no interests in life. His brother gets killed and Dolph **tries** → **attempts** to figure things out. The character is just plain stupid and stumbles around aimlessly. Pointless. |
| **Negative** → **Negative** Significant increase in FIM ($\lambda_{max} =$**1.56**) | Probably the **best** Dolph film ever. There's **everything** you'd want or expect here. **Spend** your time. Dolph portrays a miserable cop with lots of interests in life. His brother gets killed and Dolph attempts to figure things out. The character is just plain **amazing**. |

Table 5: Perturbing a difficult example on the IMDb dataset: The first row represents original sentence. Unlike easy examples, difficult examples tend to have a mixture of positive and negative traits. Furthermore, a minor perturbation like removal of a sentence (row 1) or substitution of a word (row 2) causes a significant drop in FIM eigenvalue. The small FIM score indicates that the perturbed review is not difficult for the classifier.

| Perturbed sentiment | Word substitutions |
|---|---|
| **Negative** $\rightarrow$ **Negative difficult example** ($\lambda_{max}$ =**5.47**) | It really impresses me that it got made. The director/writer/actor must be really charismatic in reality. I can think of no other way itd pass script stage. What I want you to consider is this...while watching the films I was feeling sorry for the actors. It felt like being in a stand up comedy club where the guy is dying on his feet and your sitting there, not enjoying it, just feeling really bad for him coz hes of trying. Id really like to know what the budget is, guess it must have been low as the film quality is really poor. I want to write 'the jokes didn't appeal to me'. but the reality is for them to appeal to you, you'd have to be the man who wrote them. or a retard. So imagine that in script form...and this guy got THAT green lit. Thats impressive isn't it? |
| **Negative** $\rightarrow$ **Negative Significant change in FIM** ($\lambda_{max}$ =**0.640**) | It really impresses me that it got made. The director/writer/actor must be really charismatic in reality. I can think of no other way itd pass script stage. What I want you to consider is this...while watching the films I was feeling sorry for the actors. It felt like being in a stand up comedy club where the guy is dying on his feet and your sitting there, not enjoying it, just feeling really bad for him coz hes of trying. Id really like to know what the budget is, guess it must have been low as the film quality is really poor. I want to write 'the jokes didn't appeal to me'. but the reality is for them to appeal to you, you'd have to be the man who wrote them. or a retard. So imagine that in script form...and this guy got THAT green lit. **Thats impressive isn't it?** |
| **Negative** $\rightarrow$ **Negative Significant change in FIM** ($\lambda_{max}$ =**0.445**) | It really impresses me that it got made. The director/writer/actor must be really charismatic in reality. I can think of no other way itd pass script stage. What I want you to consider is this...while watching the films I was feeling sorry for the actors. It felt like being in a stand up comedy club where the guy is dying on his feet and your sitting there, not enjoying it, just feeling really bad for him coz hes of trying. Id really like to know what the budget is, guess it must have been low as the film quality is really poor. I want to write 'the jokes didn't appeal to me'. but the reality is for them to appeal to you, you'd have to be the man who wrote them. or a retard. So imagine that in script form...and this guy got THAT green lit. Thats **impressive** $\rightarrow$ **weird** isn't it? |

Table 6: BERT: Difficult Examples change sentiment with a single word substituted. Easy examples, however, retain positive sentiment despite multiple substitutions of positive words with negative words.

| Perturbed sentiment | Word substitutions |
|---|---|
| **Positive** $\rightarrow$ **Negative difficult example** ($\lambda_{max}$ =**0.78**) | OK, I kinda like the idea of this movie. I'm in the age demographic, and I kinda identify with some of the stories. Even the sometimes tacky and meaningful dialogue seems realistic, and in a different movie would have been forgivable.

I'm trying as hard as possible not to trash this movie like the others did, but it's easy when the filmmakers were trying very hard.

The editing in this movie is terrific! Possibly the **best** $\rightarrow$ **worst** editing I've ever seen in a movie! There are things that you don't have to go to film school to learn, leaning good editing is not one of them, but identifying a bad one is.

Also, the shot... Oh my God the shots, just fantastic! I can't even go into the details, but we sometimes just see random things popping up, and that, in conjunction with the editing will give you the most exhilirating film viewing experience.

This movie being made on low or no budget with 4 cast and crew is an excuse also. I've seen short films on youtube with a lot less artistic integrity! ... |
| **Positive** $\rightarrow$ **Positive easy example** ($\lambda_{max}$ =**0.55**) | This is the **best and most** original show seen in years. The more I watch it the more I **fall in love with** $\rightarrow$ **hate** it. The cast is **excellent** $\rightarrow$ **terrible** , the writing is **great** $\rightarrow$ **bad**. I personally **loved** $\rightarrow$ **hated** every character. However, there is a character for everyone as there is a good mix of personalities and backgrounds just like in real life. I believe ABC has done a great service to the writers, actors and to the potential audience of this show, to cancel so quickly and not advertise it enough nor give it a real chance to gain a following. There are so few shows I watch anymore as most TV is awful . This show in my opinion was right down there with my favorites Greys Anatomy and Brothers and Sisters. In fact I think the same audience for Brothers and Sisters would hate this show if they even knew about it. |

Table 7: Counterfactual perturbations cause reduction in difficulty: The original example is more difficult than the perturbed counterfactual example because of strong giveaway words like "amazing". The subtle changes to make the sentence positive like changing the amount or the rating almost makes no difference. Instead, the use of a strong word "amazing" makes the sentence extremely easy for the model to classify as positive. The heavy reliance on a single word for the positive example makes this much easier to classify than the original sentence which used the word "odd" (a word that is not necessarily negative) as a negative sentiment.

| Perturbed sentiment | Word substitutions |
|---|---|
| **Negative** → **Negative difficult example** ($\lambda_{max} =$**3.42**) | Definitely an **odd** debut for Michael Madsen. Madsen plays Cecil Moe, an alcoholic family man whose life is crumbling all around him. Cecil grabs a phone book, looks up the name of a preacher, and calls him in the middle of the night. He goes to the preacher's home and discusses his problems. The preacher teaches Cecil to respect the word of God and have Jesus in his heart. That makes everything all better. Ahh...if only everything in life were that easy. The fact that this "film" looks as if it was made with about **$500** certainly **doesn't** help. **1/10** |
| **Positive** → **Positive Significant change in FIM** ($\lambda_{max} =$**0.640**) | Definitely an **amazing** debut for Michael Madsen. Madsen plays Cecil Moe, an alcoholic family man whose life is crumbling all around him. Cecil grabs a phone book, looks up the name of a preacher, and calls him in the middle of the night. He goes to the preacher's home and discusses his problems. The preacher teaches Cecil to respect the word of God and have Jesus in his heart. That makes everything all better. Ahh...if only everything in life were that easy. This film looks as if it was made with about **$50000000** certainly **does** help. **10/10** |

Table 8: Difficult examples on the IMDB counterfactual dataset. Here the first row is the original sentence and the next row is the counterfactual sentence. The counterfactual sentence is easier than the original sentence. Note the original example relied more on words like "unlucky", "doesn't", and "absolutely" during classification. "unlucky" and "doesn't" are associated with more negative sentences and thus the counterfactual example is much easier for the model along with very negative words like "terrible" and "boring".

| Perturbed sentiment | Word substitutions |
|---|---|
| **Positive** → **Positive difficult example** ($\lambda_{max} =$**4.04**) | An **excellent** movie about two cops loving the same woman. One of the cop (Périer) killed her, but all the evidences seems to incriminate the other (Montand). The **unlucky** Montand **doesnt** know who is the other lover that could have killed her, and Périer doesnt know either that Montand had an affair with the girl. Montand must **absolutely** find the killer...and what a **great** ending! **Highly** recommended. |
| **Negative** → **Negative Significant change in FIM** ($\lambda_{max} =$**0.35**) | A **terrible** movie about two cops loving the same woman. One of the cop (Périer) killed her, but all the evidences seems to incriminate the other (Montand). The **unlucky** Montand **doesnt** know who is the other lover that could have killed her, and Périer doesnt know either that Montand had an affair with the girl. Montand must **absolutely** find the killer...and what a **boring** ending! I don't recommend **at** all. |

Table 9: Difficult examples on the IMDB counterfactual dataset. Here the first row is the original sentence and the next row is the counterfactual sentence. Mix of positive and negative words make the sentences difficult for the model.

| Perturbed sentiment | Word substitutions |
|---|---|
| **Positive** $\rightarrow$ **Positive** difficult example ($\lambda_{max}$ =**4.18**) | Was flipping around the TV and HBO was showing a double whammy of unbelievably **horrendous** medical conditions, so I turned to my twin sister and said, "Hey this looks like fun," - truly I love documentaries - so we started watching it. At first I thought Jonni Kennedy was a young man, but then it was explained that due to his condition, he never went through puberty, thus the high voice and smaller body. He was on a crusade to raise money for his cause. He had the most **wonderful** sense of **humor** combined with a **beautiful** sense of spirituality... I cried, watched some more, laughed, got up to get another Kleenex, then cried some more. Once Jonni Kennedy's "time was up" he flew to heaven to be with the angels. He was more than ready; he had learned his lessons from this life and he was free. I **highly recommend** this. If you do not fall in love with this guy, you have no heart. |
| **Negative** $\rightarrow$ **Negative** Significant change in FIM ($\lambda_{max}$ =**2.45**) | Was flipping around the TV and HBO was showing a double whammy of unbelievably **horrendous** medical conditions, so I turned to my twin sister and said, "Hey this looks like fun," - truly I love documentaries - so we started watching it. At first I thought Jonni Kennedy was a young man, but then it was explained that due to his condition, he never went through puberty, thus the high voice and smaller body. He was on a crusade to raise money for his cause. He had the **worst** sense of **humor** combined with an ugly sense of spirituality... I nodded off, watched some more, snoozed, got up to get a coffee, then snoozed some more. Once Jonni Kennedy's "time was up" he flew to heaven to be with the angels. He was more than ready; he had learned his lessons from this life and he was free. I **highly recommend** you don't watch this. If you do not fall asleep within the first ten minutes, you have no taste. |

Table 10: Difficult examples on the IMDB contrast dataset. Here the first row is the original sentence and the next row is the contrast set sentence. The contrast set sentence is easier than the original sentence. The difficulty in the first sentence is due to words like "irresponsible" and "sloppy". Thus the negative sentence is much easier for the model.

| Perturbed sentiment | Word substitutions |
|---|---|
| **Positive** → **Positive** difficult example ($\lambda_{max}$ =**3.26**) | Here's another film that doesn't really need much of a recommendation. It's a classic comedy, very **funny** and **entertaining** and which, of course, ultimately inspired a successful television series which many would say was even better (I **enjoy** both, personally). \<br /\>\<br /\>For some, it's hard to warm up to Jack Lemmon and Walter Matthau as Felix Unger and Oscar Madison when they were were weaned on the TV show starring Tony Randall and Jack Klugman (or perhaps vice versa). But what we've got there in both cases are four good actors who in real life seemed so much like their film counterparts that they managed to make these characterizations their own. It's Neil Simon's humorous material that's key, and where the laughs really originate from.\<br /\>\<br /\>For those who have somehow never heard of THE ODD COUPLE, it's the story of a neurotic and fussy neat-freak (Lemmon) who is thrown out of a 12-year marriage by his long-suffering wife and takes up residence in the Manhattan apartment of his **sloppy** and totally **irresponsible** buddy (Matthau). Pitting these two unlikely roommates together within the same four walls makes for some hugely funny predicaments. |
| **Negative** → **Negative** Significant change in FIM ($\lambda_{max}$ =**0.32**) | Here's another film that really needs a recommendation to watch. It's a travesty, **unfunny** and which, of course, ultimately inspired a unsuccessful television series which many would say was even **worse** (I hated both, personally). \<br /\>\<br /\>For some, it's hard to warm up to Jack Lemmon and Walter Matthau as Felix Unger and Oscar Madison when they were were weaned on the TV show starring Tony Randall and Jack Klugman (or perhaps vice versa). I am no exception. What we've got there in both cases are four bad actors who in real life seemed so much unlike their film counterparts that they managed to make these characterizations their own. It's Neil Simon's material that's the worst, and where the fails really originate from.\<br /\>\<br /\>For those who have somehow never heard of THE ODD COUPLE, it's the story of a neurotic and fussy neat-freak (Lemmon) who is thrown out of a 12-year marriage by his long-suffering wife and takes up residence in the Manhattan apartment of his **sloppy** and totally **irresponsible** buddy (Matthau). Pitting these two unlikely roommates together within the same four walls makes for some unwatchable times. |

Table 11: Easy and difficult examples in Contrast Sets

| Perturbed sentiment | Word substitutions |
|---|---|
| **Negative** → **Negative** easy example ($\lambda_{max}$ =**0.034**) | This is a **pathetic** → **excellent** political satire. No wonder why it was largely ignored in the U.S.: it ridicules our foreign policy and misrepresents what it really is.

Another **bad** → **good** film from this era, Rendition, was however totally dismissed simply because it showed, accurately, that the U.S. is a war machine bent on torturing, murdering, and maiming civilians in its quest for total world domination.

A **factually incorrect** → **correct**, **bad** → **good** acting, some big stars (John Cusack, Ben Kingsley, Marisa Tomei anyone?) and some scenes of hilarity but they couldn't have made this movie a hit. Thankfully, Americans don't like to hear misrepresentations about anyone, even if they are complicit in mass murder. |
| **Positive** → **Negative** difficult example ($\lambda_{max}$ =**4.132**) | I'm a big Porsche fan, and the car was the best star in this film.

Haim, the drug abusing child star of the 80's is **amazing** → **horrible, excellent** as per usual, and commenting on back up from minor characters/actors would be pointless; **needless** to say they were all above average. It's a cool movie as a trip down memory lane into the 80's - with some weird clothes, some good shots of the Colorado backdrop and a very mind stimulating plot.

All in all, please watch this unless you hate 80's movies, Corey Haim, or unlike myself, hate old school Porsches (this one in particular looks great) because life's too short to instead watch **crappy** movies. |
| **Negative** → **Positive** difficult example ($\lambda_{max}$ =**3.98**) | The first film was an okay one, and it is nowhere as good as the **wonderful animated** classic which I found more poignant and endearing. This sequel is not just inferior, its really bad. Yes the slapstick is too much, the script has its weak spots and the plot is a tad uninspired. Yeah probably the dogs are very cute here, but Eric Idle is dumb as a cow. The film is a pain to look at with any cinematography and eye hurting costumes(especially Cruella's), and the music is sleepy. The acting is mostly very bad, Ioan Gruffudd is **appalling** → **terrible, shocking, excellent, extraordinary** and Gerard Depardieu while he has given better performances has little fun as Cruella's accomplice. But the best asset, as it was with the first film, is the amazing Glenn Close in a deliciously over-the-top performance as Cruella, even more evil than she was previously. Overall, poor. 3/10 Bethany Cox |

Table 12: Difficult examples for BERT. Sensitive to single word substitutions. In example 1, each word was substituted one at a time.

| Perturbed sentiment | Word substitutions |
| --- | --- |
| **Positive** → **Negative** difficult example ($\lambda_{max}$ =**0.60**) | "The director tries to be Quentin Tarantino, the screenwriters try to be Tennessee Williams, Deborah Kara Unger tries to be Faye Dunaway, the late James Coburn tries to be Orson Welles, Michael Rooker tries to be Gene Hackman, Mary Tyler Moore tries to be Faye Dunaway (older version), Cameron Diaz tries to get out of the frame as quickly as she can (successfully), don't ask about Joanna Going. And they actually pull it off. Eric Stoltz and James Spader try to present their joy with this stuff. It delivers **thoughtful** → **thoughtless** , **meaningful** → **meaningless** dialog and very little action.\<br /\>\<br /\>Tulsa is a town with beautiful elevator lobbies, an art deco church by Bruce Goff and a lovely, sprawling mansion by Frank Lloyd Wright. Visit Tulsa, consider watching this movie. It doesn't do the location justice, but still worth it.". |
| **Negative** → **Positive** difficult example ($\lambda_{max}$ =**0.58**) | "Many people here say that this show is for kids only. Hm, when I was a kid (approximately 7-9 years old) I watched this show first. It was **disgusting** → **okay** for me. I talked with other kids about this and, sure, other shows and know what? This was the measure of disguise, whenever we wanted to emphasize something's silliness (either on TV or anything else) we said 'Uh, just like Power Rangers' and laughed. \<br /\>\<br /\>And before visiting this site I could not imagine that there actually are fans of MMPR. It was so strange for me that I decided to watch it again and try to understand why people like it. I did not enjoy that viewing. But it dawned upon me: maybe I have not enough imagination? It may be. However this argument is not sufficient for me to rate it more than 1 star." |

