# OpenReview forum: "Geometry matters: Exploring language examples at the decision boundary"
_ICLR.cc/2021/Conference — Reject_

### Official Review · AnonReviewer3 · 2020-10-28

**Rating:** 5
**Confidence:** 2

**Review:**

This paper proposes an analysis technique for studying the 'difficulty' of a pair of test dataset examples in NLP. The setup proposed by past work on Contrast sets and Counterfactual Examples (Gardner et al, 2020 and Kaushik et al 2020 respectively) is to manually construct two dataset examples (x,y) with different labels y, while the inputs x differ only minimally. This paper argues to compute the measure of a contrast / counterfactual example pair by extracting the largest Eigenvalue of a matrix (defined in part using the Fisher Information Matrix).

Strengths:
* the idea of using the Fisher Information Matrix to categorize the difficulty of (contrast) examples in NLP seems new to this reviewer and well motivated.
* The paper argues that many of the existed (hand-crafted) perturbations are not difficult enough for models because they lie far away from the decision boundary, which could possibly lead to insights about how to build better contrast sets later on.

Weaknesses and questions:
* To this reviewer, it is unclear why this paper leads to a different conclusion than that of Gardner et al, 2020 or Kaushik et al 2020. It seems like those authors were able to mine 'difficult' examples where 'difficult' could be defined as resulting in a flipped prediction -- at least in their experimental setup. It is unclear why if that is the case from the original papers, why the FIM eigenvalue doesn't increase -- but perhaps I'm missing something :) It would make this paper much stronger if the FIM metric was also compared with metrics proposed by those other papers, such as difference in accuracy.
* How do you define $\eta$ for BERT-style models? Particularly since the representations that these models learn about what a sentence is change quite a bit during finetuning.
* Perhaps this is more for future work, but to this reviewer, it is not clear what we learn about the insights of fragility of NLP models from this direction. The methodology would be a lot stronger if we could use this technique to sample examples that are quantifiably difficult for existing models, e.g.

* Nie et al 2019 (Adversarial NLI)
* Zellers et al 2019, HellaSwag: Can a Machine Really Finish Your Sentence?

Overall: at least to this reviewer, this work seems promising but there are a few open questions about how it works and how insightful it is. I would be happy to raise my score if those questions were answered appropriately.

----

Update: thanks for the response! I read over the updated draft also but I'm still not sure what insights we learn about the fragility of NLP models under this evaluation paradigm. For that reason I'm still confused as to whether this paradigm is better or worse than the original approaches of Gardner et al / Kaushik et al 2020, and so I'd like to stick to my score.

---

> ### Author Response · Authors · 2020-11-18
> **More empirical experiments added to show relationship with classifier accuracy**
>
> The different conclusion from Gardner et al: We thank the reviewer for this astute observation. Truthfully, we were pretty puzzled by the result on contrast sets. One potential reason for a different conclusion than Gardner et al is that when sentences are tweaked, the type of perturbation performed can make the sentence simpler. We do extensive experiments on three other datasets to show the relationship between accuracy and FIM. We randomly sample words from the training set and show the relationship with accuracy. We also perturb sentences along the direction of the eigenvector and show the relationship with accuracy. In some cases, it can be seen the model performance drops as much as 50% on the test set when high FIM values are perturbed.
>
> In experiments 2 and 3 we show random sampling based on FIM value can provide insights on difficult examples. This method can thus be used to sample difficult examples and we would release this code as well to sample examples susceptible to perturbation, i.e to find difficult examples for any dataset. Dynabench (Adversarial NLI) are things we plan to explore in the future. We noticed Dynabench platform releases different rounds for each task and it would be worth exploring how the FIM value changes for each round of the task.

---

### Official Review · AnonReviewer1 · 2020-10-28
**Interesting idea but lacking concrete empirical/theoretical results**

**Rating:** 3
**Confidence:** 4

**Review:**

**Update after author response:**
I went over the author response and have had a chance to carefully evaluate the updated draft. I appreciate that the authors have taken the time to address two of my comments but I believe major concerns still remain unaddressed, so my evaluation remains unchanged.

1. I appreciate the time taken to conduct the fastText experiments, would be helpful to show this on BERT (which is arguably your strongest model) as well.

2. Per my reading it appears that you've shown how models behave if the replacements are sampled at random. I appreciate the time taken to show that. I believe this could be made better with a concrete discussion of the same.

Cons 3, 4, 5, and Additional comments 1,2: I do not think these have been convincingly addressed. You point to Dynabench while suggesting that this would eventually lead to better evaluation sets but as it is shown in [1] and [2], "difficult" evaluation sets that are tied to one model are not necessarily difficult for other models. If the idea is to identify model's vulnerabilities and not use these as common evaluation sets, then this is more or less the same as generating adversarial examples (but called "difficult" examples, in which case this would be better presented as an application/extension of Zhao et al to identify adversarial examples in NLP). If the idea is to only use these examples for evaluation purposes, then the comparison to counterfactually augmented data (which you show does not include "difficult" examples per your definition) does not make sense since that is meant solely for augmenting training sets. Discussion in Section 4 and onwards is shallow and often unclear. That is primarily where I believe the exposition can be improved significantly. For instance, Section 4 says FIM values capture resilience to linguistic perturbations but that has not been discussed in any of the paragraphs that follow. Section 5.1 ends in a sentence that says, "By repeating this process multiple times, more robust classifiers can be created", and that is not discussed any further as to why you think that would be the case. It is also not supported by any theory or empirical results presented in your prior or updated draft.

[1] Eric Wallace, Pedro Rodriguez, Shi Feng, Ikuya Yamada, and Jordan Boyd-Graber. "Trick me if you can: Human-in-the-loop generation of adversarial examples for question answering." In TACL.

[2] Max Bartolo, Alastair Roberts, Johannes Welbl, Sebastian Riedel, and Pontus Stenetorp. "Beat the AI: Investigating Adversarial Human Annotations for Reading Comprehension." In TACL.

---------------------------------------------------------

Summary:

The paper proposes using the largest eigenvalue of the Fisher information metric (previously studied to identify adversarial perturbations in computer vision) as a measure to construct "difficult examples" for the IMDb sentiment analysis task. The authors suggest that one may perform such perturbations by replacing "important" words in the reviews. They further analyze those examples comparing them to counterfactually revised data (CRD; Kaushik et. al, 2020) and contrast sets (Gardner et al., 2020) and show that based on the FIM criteria, CRD and contrast sets are not too different from original reviews in the IMDb dataset.

Pros:
- This paper describes a simple method for understanding difficulty in terms of ease-of-perturbation.
- The authors find that, interestingly, sentiment analysis datasets generated by counterfactually revising data are not "difficult" for classifiers.

Cons:
- While the idea of using FIM to identify examples closer to a classifier’s decision boundary is interesting, it is a new application to NLP even though the idea has been discussed extensively in Zhao et al. (2019). However, the paper doesn't show why these examples may be useful. The paper reports no empirical results or analysis thereof to understand the practical efficacy of these "difficult examples" while discussing only the changes in eigenvalues.
- It is not clear as to how generalizable this approach is. For instance, the paper uses a replacement strategy based on substituting synonyms, antonyms, and certain kinds of noun phrases (actress names etc.)  which may not directly translate to other tasks such as question answering, news classification, VQA, or even NLI.
- The paper makes many philosophical claims without theoretical or empirical justification. At one point it suggests that FIM captures resilience to linguistic perturbations but provides no backing for the claim. The paper further suggests that evaluation sets should increase FIM but provides no theoretical/empirical justification for the same. At another instance it says, “[t]ransfer learned models like BERT capture rich semantic structure … and tend to rely on semantically relevant words for classifying movie reviews” without citing any prior work that supports this claim.
- The approach is also very model specific. For one model some data points may turn out to be not "difficult" but for a different model they may, which casts further doubt on the general applicability of this method in this setting. Note Zhao et al use it to identify adversarial examples, which by definition are specific to a model. Furthermore, if we go by the authors' suggestion that evaluation sets must increase FIM, then by using such "difficult" examples for evaluation we end up discriminating (by design) against the model that was used to identify them, and unintentionally favor other models, leading to a potentially flawed comparison. I don't think that it is a desirable characteristic of an evaluation set.
- Particularly given the lack of empirical evidence showing the benefits of "difficult examples" identified by this approach, the core claim of the paper is that examples generated by prior work are not "difficult" in terms of the discussed metric and perhaps they need to be so. However, the motivation expressed in Kaushik et al. is not to create "difficult" examples for training or evaluation, but to learn better models by intervening on causal variables of interest. A good counterfactual example generated by their approach would be one that modifies only sentiment-related variables and leave others intact (intervene on causal variables to d-separate labels from the spuriously correlated variables, see [1] for a detailed explanation), and it has no relation to any particular classifier. I agree that Gardner et al. do discuss their motivation as to construct examples closer to the decision boundary.
- The writing can be significantly improved.

Additional comments:
- It is also not clear to me how replacements are sampled, it would be good to provide details on that.
- It would be nice to see how well models perform in-sample and out of domain when trained on "difficult examples" and how models trained on CAD perform on previously identified "difficult examples" in test sets. Furthermore, are there overlaps between such examples if FIM is calculated with respect to a model trained on original data alone vs. one trained on CAD?

Missing references:
- Please cite the IMDb dataset: "Maas, Andrew, et al. Learning Word Vectors for Sentiment Analysis. Proceedings of the 49th Annual Meeting of the Association for Computational Linguistics: Human Language Technologies. 2011."
- Prior work building on Gardner et al. and Kaushik et al. demonstrating the efficacy of their work:
Teney, D., Abbasnedjad, E., & Hengel, A. V. D. (2020). Learning what makes a difference from counterfactual examples and gradient supervision. arXiv preprint arXiv:2004.09034.
- There are also a lot of arxiv citations for papers that have been peer reviewed.

Errata:
- Page 2, line 3: “BERTby” -> BERT by
- Page 2, second paragraph, line 1: Fisher information metric has already been defined in the previous paragraph so you may refer to it just as FIM over here.
- Page 3: Glockner et al. should be \citep and not \citet. Same for Rajpurkar et al., Recht et al., and Beery et al.
- Page 3: “learning byMitchell et al. (2018)” -> learning by Mitchell et al. (2018)


[1] Kaushik, D., Setlur, A., Hovy, E., & Lipton, Z. C. (2020). Explaining The Efficacy of Counterfactually-Augmented Data. arXiv preprint arXiv:2010.02114.

---

> ### Author Response · Authors · 2020-11-18
> **Added concrete theoretical and empirical results**
>
> Thank you for your review. We have now added concrete empirical results across three other datasets of different sizes and number of output labels  (YelpReviewPolarity, AGNEWS, and SogouNews). Please see the RebuttalExperiments.pdf in the supplementary.
>
> *  We have additional experiments to mine a set of difficult examples for the model. We report empirical results random substitutions of words from the training set for both easy and difficult examples.  Our results were consistent across the three datasets. We provide results on accuracy in figure 1 of the supplementary. Drop in accuracy as much as 50 percent on this set when the examples in the data are perturbed. By studying the high fim examples, more robust datasets can be created in multiple iterations, similar to dynabench (https://dynabench.org/about).
> *  We also report empirical results of perturbation along the direction of the eigenvector for random examples and show that lower perturbation is necessary for examples with high fim eigenvalue to flip the classifier label.
> *  We also show how difficult and easy examples respond to perturbation across three datasets by just sampling these examples based on fim eigenvalue. (This is our sampling strategy as can be seen in figure 1). For the other examples in the supplementary, we select random examples to show the relationship with FIM.
> * By mining the difficult examples for models like BERT, fastext and CNN and taking the common ones, we can discover examples that are difficult across multiple models.
>
> Thank you again for pointing out the reference issues and edits. Do you have any particular feedback about our writing? We would incorporate those changes in the subsequent draft.

---

### Official Review · AnonReviewer4 · 2020-10-30
**interesting observation, but the method is used and evaluated for the wrong purpose**

**Rating:** 4
**Confidence:** 5

**Review:**

## Summary
The authors argue that we should evaluate the robustness of NLP models near their decision boundaries, and argue that contrast sets and counterfactual examples cannot fullfill this purpose. The authors propose to find examples near the decision boundary using the largest eigenvalue of the Fisher information matrix, arguing that this value gives us a sense of how stable the model prediction is near the input. To verify that FIM can identify examples where the prediction is unstable, and perturbation leads to larger prediction change, the authors use some heuristic adversarial attacks: first identify tokens to replace using integrated gradients, then replace the tokens with synonyms (to confirm the prediction is sensitive) or antonyms (to confirm that the prediction is insensitive).

## Detailed comments
IG-based perturbation: I was initially confused by how the authors plan to use IG for perturbation, since IG gives a scalar score for each word and does not tell us a direction for the word embedding to move along. Then I realized that IG is only used to select which token to perturb, and the actual perturbation is done by replacing words with their synonyms/antonyms. I think it’d help to clarify this point.

Framing: the largest eigenvalue of FIM tells us how easy it is to change the output probability distribution of the model by perturbing the input, but we don’t know if the groundtruth label of the example will change after perturbation. In other words, in the general case, whether or not the perturbation leads to adversarial examples still requires human verification. In this paper the authors get around doing human verification by replacing words with only synonyms or antonyms. But I think it would help reinforce the message to make this point clearer. For example, instead of saying “ and these examples are most susceptible to perturbations” (from second paragraph of section 4.1), it might be better to say “the model prediction on these examples are most susceptible to perturbations to the input”. I’d argue against calling examples selected by FIM “difficult” and “easy”, maybe “unstable” and “stable” is better. Keep in mind that FIM is computed specifically for a model, and the largest eigenvalue of FIM describes a property of the model’s induced distribution near that example, not the example itself.

Why do we need FIM? This paper did not introduce a new perturbation, it introduced a way to identify examples that have high potential to be perturbed into adversarial examples. Under this framing, I’m not sure that the proposed method is evaluated against proper baselines. Importantly, the authors motivate this work by first observing that contrast sets and counterfactual examples do not really evaluate the models near their decision boundaries. That is probably a fair assessment, but those work have a different goal in mind. Both contrast sets and counterfactual examples ask human annotators for small changes that significantly change the _groundtruth label_ of the example. Maybe these edited examples are not near the decision boundary, that’s because they are not designed to be; being close to the decision boundary is purely incidental. I think this is an important distinction between contrast set/counterfactual examples and gradient-based adversarial attacks such as [HotFlip](https://arxiv.org/abs/1712.06751) and the human-in-the-loop counterpart such as [TrickMe](https://arxiv.org/abs/1712.06751).

The authors verify that the examples selected by FIM are indeed “difficult” (again, this is not the best term) using several heuristic-based attack (replacement with synonym, antonym, and replacement with actress names). I don’t think this is a convincing evaluation of FIM as a difficulty measure: one can simply try these attacks and find the same set of “difficult” examples. A more valid baseline is perhaps selecting examples by prediction confidence.

I think whether “models should be evaluated near decision boundary” is debatable. Yes, it’s likely easier to create adversarial examples near the decision boundary. But considering that we cannot yet claim that robustness in NLP is solved for the “easier” examples, I’m not sure focusing on examples near the decision boundary is the right objective right now.

Figure 1(a): no details were provided as to how the figure is generated for IMDb examples. How did you reduce the dimensionality of the decision boundary to 2D? How did you pick the examples? What does the green band signify? Are these real examples & real models or is this figure illustrative?

Figure 1(c): are colors (blue vs red) indicative of the sign of the eigenvalue? It looks like the direction of the corresponding eigenvectors are only visualized for red points, how are they selected?

---

> ### Author Response · Authors · 2020-11-18
> **Misunderstanding about decision boundaries.**
>
> Thank you for your comments. Finally, the examples should be useful to the model in some way, Consider the scenario of active learning or semi-supervised learning, where if you were to pay x dollars for each labeled example (generated by a human being). If you were to generate examples that are far from the decision boundary, the classifier would still be able to classify most of them, without being able to affect classifier performance, and we would end up paying a huge cost.
>
> Note that we are not arguing against humans in the loop systems. What we are claiming is that these approaches should be augmented with FIM in order to validate if the examples generated are actually novel for the classifier and whether the perturbations are actually small in embedding space. We validate this across 3 new datasets and tasks in the supplementary.
>
> * Evaluating classifier performance by using classifier confidence is risky. As Gardner et al themselves argue in their paper, because of systematic gaps or shortcut learning (as we mentioned earlier) in our paper, the classifier might be extremely confident in its incorrect prediction. It's important to study the behavior in a local epsilon ball around the decision boundary. By studying the local KL divergence over y, we provide a theoretically motivated method for the same.
>
> * Not sure what the reviewer means by models should not be evaluated near the decision boundary. Consider the case of logistic regression. Here, a perpendicular distance from the plane separating the two classes is what decides the label of each data point. For points far away from the decision boundary, no local/non-gratuitous (counterfactual examples) perturbation can make the data point cross the decision boundary and switch labels, and it’s not interesting to study their behavior.
>
>
> 1. This figure a is illustrative. Whereas figures b and c are from a two-component Gaussian Mixture model where points were sampled and colored by the eigenvalue of the FIM (figure c)
> 2. The red points are the points close to the decision boundary. The points are colored by the eigenvalue of the FIM.

---

### Official Review · AnonReviewer2 · 2020-10-31
**Potential of 1st Eigenvalue of Fisher Information Metric to identify difficult examples, More experimental results needed**

**Rating:** 5
**Confidence:** 4

**Review:**

Summary:
This paper defines a new metric to quantify the difficulty of an example in language classifiers: the largest eigenvalue of the Fisher Information Metric (FIM).
They show a geometric interpretation of the FIM, the greater the largest eigenvalues of the FIM, the closer the example is to the decision boundary. The authors illustrate this with a toy example. They then show how perturbations of examples with high eigenvalues for FIM, e.g., substitutions of one or more words, can flip the output of a classifier. Whereas, examples with lower eigenvalues are not as vulnerable to substitutions. The paper also argues that previous work on adversarial examples using contrast sets and counterfactual examples, are in fact not as vulnerable to substitutions, measured by the FIM.

Strengths:
- Development of a quantifiable metric to identify difficult examples. It would indeed be great to have such a metric, and help model developers to identify examples that are especially vulnerable to small perturbations. This in turn can help in developing more robust models. Therefore, I think there is potential for broad impact of this work.
- As far as I know, the idea of using eigendecomposition of FIM to differentiate between easy and difficult examples for NLP applications is novel. It has been used to generate adversarial examples in computer vision. Though I think the application to NLP is a strong independent contribution. Assuming the claim can be established more definitively, the community can report results on thus identified difficult examples thus allowing us to inspect the vulnerabilities of any model.
- The potential geometric interpretation of FIM is appealing. If this can be shown to hold given the nonlinearities in deep learning models this would be a very impactful result.
Weaknesses:
1. More empirical results are needed. Alternately, theoretical proofs may be used to substantiate the claims.
a. The idea that examples associated with large eigenvalues for FIM are more sensitive to perturbation is illustrated on a few select examples. However, it would be great to see this hold across the dataset. One way to show this might be to perturb all or some subset of examples in the test set, and plot the number of perturbations needed to flip the classifier output against largest eigenvalue of FIM.
b. The reasoning that high eigenvalues correspond to points near the decision boundary is made based on an illustration on a linearly separable toy example. It is not clear this will extend to deep learning models with more complex decision boundaries.
c. The only case against contrastive sets and counterfactual examples is that they dont have a high eigenvalue for the FIM. However, as this paper does not definitively establish yet that high FIM values is always correlated with difficulty in classification/sensitivity to perturbation, the argument against competing methods is insufficient.
d. Since the use of eigenvalues as FIM as a measure of example difficult is generally applicable across several NLP tasks, the authors can make a stronger case by adding more datasets and tasks to the experimental results.
2. Related work should be explained in more detail. A case is made comparing the proposed metric against contrast set and counterfactual examples (recent works). These should be described in more detail so paper is self-contained.
Similarly, prior work around eigendecomposition of FIM should be given due credit. Zhao (2019) is mentioned in the introduction. However, the derivation in the methods section should be duly attributed as well.

Recommendation:
My recommendation is to reject the paper at this time. While I think there is great potential in the idea of using eigenvalues of the FIM to identify difficult examples, I think the experimental results are not sufficient to support the case as described above.

I encourage authors to make a stronger case in a future revision, by addressing the following points:
- On any existing task/model, can it be shown that examples that are erroneously classified typically have high eigenvalues for FIM.
- Can the result be demonstrated on more than one dataset/task?
- Can the result be demonstrated on more than a handful of examples for a task. e.g., by reporting average eigenvalues vs number of perturbations described above.
- Can the geometric interpretation be illustrated on more complex decision boundaries?

---

> ### Author Response · Authors · 2020-11-18
> **More datasets and tasks and programmatic evaluation**
>
> Thank you for your insightful comments. We realized that experimental evaluation on multiple datasets was going to make our case stronger and we have now addressed most of your concerns through these experiments.
>
> Specifically:
> *   Because of shortcut learning, examples can be erroneously classified but may not have high eigenvalues of FIM.
> *   We have demonstrated this on three other datasets and tasks.
> *   We report the average eigenvalue and number of perturbations (word flips). We also report based on the perturbation strength of the l2 norm along the eigenvector.
>
> We will also update the revised document to expand the related work and highlight the work of Zhao (2019) for the derivation.

---

### Author Response · Authors · 2020-11-18
**More experiments and evaluations added**

Thank you all for your insightful comments. This actually inspired us to do new sets of experiments to validate our claims. We validated our claims on linguistic resilience by performing additional experiments on programmatic sentence perturbations in both embedding space and by token substitution. All our experiments are in the RebuttalExperiments.pdf in the supplementary. In figure 1, we mine difficult examples and present accuracy results of a classifier on the same.

We also do extensive experiments on three other datasets YelpReviewPolarity, AGNEWS, and SogouNews since there is now a convenient way to access them through torchtext’s newest release. A wide range of datasets of different sizes with different numbers of output classes should hopefully make for a stronger case for our approach. We hope to finish some more dataset experiments over this week and update the results.

EDIT: We now have evaluations for 5 datasets (YelpReviewPolarity, AGNEWS, SogouNews, YelpReviewFull and YahooAnswers) and have edited the main paper to include the details of the analysis and experiments as suggested by the reviewers. Thank you all for your suggestions about these experiments.

---

### Decision · Program_Chairs · 2021-01-07
**Final Decision**

**Decision:**

Reject

**Comment:**

This paper presents the method of using Fisher information matrix values to identify examples near the decision boundary for a model, and proposes to preferentially use these examples in evaluation.

Pros:
- Reviewers found this use of FIM values to be novel and interesting.
- The paper presents fairly extensive results demonstrating the properties of examples selected and perturbed under the proposed methods.

Cons:
- One of the main aims of this paper is to promote the use of examples selected in this way as evaluation sets. The method relies on values estimated from a specific model to select difficult examples, so this raises a fairly serious objection: If our goal in evaluation is to produce a fair comparison of two models, how do we choose which model we should use to select the test examples?

Reviewers were fairly unanimous in the objection that they raised, and while there was substantial discussion both with the authors and among reviewers, no reviewer was satisfied that the authors took this concern seriously or offered a clear resolution.